# Elevated Plasma Bioactive Adrenomedullin and Mortality in Cardiogenic Shock: Results from the OptimaCC Trial

**DOI:** 10.3390/jcm10194512

**Published:** 2021-09-29

**Authors:** Koji Takagi, Bruno Levy, Antoine Kimmoun, Òscar Miró, Kévin Duarte, Ayu Asakage, Alice Blet, Benjamin Deniau, Janin Schulte, Oliver Hartmann, Gad Cotter, Beth A Davison, Etienne Gayat, Alexandre Mebazaa

**Affiliations:** 1Inserm UMR-S 942, Cardiovascular Markers in Stress Conditions (MASCOT), Université de Paris, 75010 Paris, France; kojitakagi@nms.ac.jp (K.T.); akimmoun@gmail.com (A.K.); aliceblet@gmail.com (A.B.); benjdeniau@gmail.com (B.D.); gadcotter@momentum-research.com (G.C.); bethdavison@momentum-research.com (B.A.D.); etienne.gayat@aphp.fr (E.G.); 2Momentum Research, Inc., Chapel Hill, NC 27517, USA; 3Service de Médecine Intensive et Réanimation Brabois, CHRU de Nancy, 54511 Vandœuvre-lès-Nancy, France; blevy5463@gmail.com; 4U1116, Défaillance Circulatoire Aigue et Chronique, Faculté de Médecine de Nancy, 54500 Vandœuvre-lès-Nancy, France; kevin.duarte@inserm.fr; 5Université de Lorraine, CS25233, CEDEX, 54052 Nancy, France; 6Emergency Department, Hospital Clínic, 08036 Barcelona, Catalonia, Spain; OMIRO@clinic.cat; 7IDIBAPS (Institut d’Investigacions Biomèdiques August Pi i Sunyer), 08036 Barcelona, Catalonia, Spain; 8Medical School, University of Barcelona, 08036 Barcelona, Catalonia, Spain; 9INSERM, Centre d’Investigations Cliniques Plurithématique 1433, Institut Lorrain du Cœur et des Vaisseaux, 54500 Vandœuvre-lès-Nancy, France; 10Department of Emergency and Critical Care Medicine, Yokohama City Minato Red Cross Hospital, Yokohama 2318682, Japan; a_asakage@yahoo.co.jp; 11Department of Anesthesiology, Critical Care and Burn Center, Lariboisière-Saint-Louis Hospitals, DMU Parabol, AP-HP Nord, University of Paris, 75010 Paris, France; 12SphingoTec, Neuendorfstraße 15A, 16761 Hennigsdorf, Germany; schulte@sphingotec.de (J.S.); hartmann@sphingotec.de (O.H.)

**Keywords:** adrenomedullin, biomarkers, cardiogenic shock, outcome

## Abstract

Aims: Bioactive adrenomedullin (bio-ADM) was recently shown to be a prognostic marker in patients with acute circulatory failure. We investigate the association of bio-ADM with organ injury, functional impairment, and survival in cardiogenic shock (CS). Methods: OptimaCC was a multicenter and randomized trial in 57 patients with CS. In this post-hoc analysis, the primary endpoint was to assess the association between bio-ADM and 30-day all-cause mortality. Secondary endpoints included adverse events and parameters of organ injury or functional impairment. Results: Bio-ADM values were higher in 30-day non-survivors than 30-day survivors at inclusion (median (interquartile range) 67.0 (54.6–142.9) pg/mL vs. 38.7 (23.8–63.6) pg/mL, *p* = 0.010), at 24 h (*p* = 0.012), and up to 48 h (*p* = 0.027). Using a bio-ADM cutoff of 53.8 pg/mL, patients with increased bio-ADM had a HR of 3.90 (95% confidence interval 1.43–10.68, *p* = 0.008) for 30-day all-cause mortality, and similar results were observed even after adjustment for severity scores. Patients with the occurrence of refractory CS had higher bio-ADM value at inclusion (90.7 (59.9–147.7) pg/mL vs. 40.7 (23.0–64.7) pg/mL *p* = 0.005). Bio-ADM values at inclusion were correlated with pulmonary vascular resistance index, estimated glomerular filtration rate, and N-terminal pro-B-type natriuretic peptide (r = 0.49, r = –0.47, and r = 0.64, respectively; *p* < 0.001). Conclusions: In CS patients, the values of bio-ADM are associated with some parameters of organ injury and functional impairment and are prognostic for the occurrence of refractory CS and 30-day mortality.

## 1. Introduction

The European Society of Cardiology and the American Heart Association defined cardiogenic shock (CS) as a state of critical end-organ hypoperfusion caused by primary cardiac dysfunction [1,2,3]. Although CS occurs in only 4–8% of patients with acute myocardial infarction, acute myocardial infarction (AMI) is the etiology of 70% of CS cases [4,5]. Early revascularization had a beneficial impact on mortality in patients with AMI; however, the mortality of CS complicating AMI remains high, approximately 50% [6,7,8]. Risk stratification for patients with CS may play an important role in initiating strict monitoring and more aggressive treatment. To date, few studies of risk stratification with biomarkers have been performed in CS patients. Adrenomedullin (ADM) is a peptide hormone consisting of 52 amino acids, which was identified from human pheochromocytoma in 1993 by Kitamura et al. [9]. Several studies showed that plasma ADM values increased in patients with hypertension [10], heart failure [11,12], AMI [11,13], pulmonary hypertension [14,15], and sepsis [16,17,18,19] and suggest that ADM was a marker of endothelial function [20]. Recently, a novel immunoassay has been developed, which directly measures bioactive ADM (bio-ADM) [21]. The latter has been shown to be a prognostic marker in patients with acute circulatory failure (e.g., septic shock, CS) [17,22]. However, the prognostic implication of bio-ADM and its relationship with the parameters of organ injury or functional impairment in patients with CS remain poorly characterized. The purpose of this study was to investigate the association of bio-ADM with organ injury, functional impairment, and survival to contribute to risk stratification and therapeutic decision-making in patients with CS.

## 2. Methods

### 2.1. Study Design

This is a post-hoc analysis of the OptimaCC trial, a multicenter and double-blinded randomized trial of 57 patients conducted in France between September 2011 and August 2016, which compared the use of epinephrine versus norepinephrine in patients with CS after AMI. The study was approved by the Nancy hospital institutional review board (ethical approval date 2011-17) and was registered on clinicaltrial.gov (NCT01367743). Written informed consent was obtained from the patients or their closest relatives. The primary outcome of the randomized trial was the change in cardiac index, and the secondary outcomes were changes in other hemodynamic variables, cardiac power index, lactate values, lactate clearance, biomarker values, and Sepsis-related Organ Failure Assessment (SOFA) evolution. A detailed description of the study design and main results have been previously published [23]. Of note, the safety monitoring board reported in 2015 a significant imbalance between the epinephrine and norepinephrine groups for the occurrence of refractory CS. This safety event led to the early termination of the study in 2016 with 57/80 patients included.

### 2.2. Participants and Inclusion, Non-Inclusion Criteria in the Princeps Study

Inclusion criteria were adult patients with a CS due to AMI treated by the percutaneous coronary intervention (PCI). CS was defined by systolic arterial pressure <90 mmHg or mean arterial pressure <65 mmHg without vasopressor or a need for vasopressors to correct hypotension, cardiac index <2.2 L/min/m^2^ in the absence of vasopressor or inotrope therapy, and at least one evidence of tissue hypoperfusion. Regarding pulmonary artery catheterization, pulmonary artery occlusion pressure had to be >15 mmHg or echocardiography had to evidence high filling pressure. Left ventricular ejection fraction had to be <40% on echocardiography without inotrope support. Non-inclusion criteria were mainly shock of other origins; immediate indication for mechanical circulatory support (MCS); cardiac arrest with early signs of cerebral anoxia; septic, toxic, and obstructive cardiomyopathy. Refractory CS was previously defined [23] as a CS with major cardiac dysfunction assessed according to echocardiography, elevated lactate value, acute deterioration of organ function despite the use of >1 µg/kg/min of epinephrine/norepinephrine or dobutamine >10 µg/kg/min and/or intra-aortic balloon support and sustained hypotension (systolic arterial pressure <90 mmHg or mean arterial pressure <65 mmHg) despite the adequate intravascular volume. 

### 2.3. Study Endpoints 

For this post-hoc analysis, the primary endpoint was to assess the association between bio-ADM at inclusion of patients and 30-day all-cause mortality. Secondary endpoints included relationships between bio-ADM and refractory CS, arrhythmias, MCS, and parameters of organ injury and functional impairment.

### 2.4. Bio-ADM Measurement

Bio-ADM measurements were achieved, blinded from the clinical data, in the laboratories of SphingoTec GmbH (Hennigsdorf, Germany) in samples collected at inclusion, 24 h, and 48 h. There were six and 12 patients with missing bio-ADM data at 24 h and 48 h, respectively. Of the patients for whom bio-ADM could not be measured at 24 h and 48 h, three patients had already died at 24 h.

### 2.5. Statistical Analyses

Analytical data are the median with 25th and 75th percentiles (median (interquartile range)) for continuous variables, whereas categorical variables are presented as numbers and percentages. Comparisons of inclusion characteristics, according to groups, were conducted by using the Mann-Whitney or Kruskal-Wallis test for continuous variables and the Fisher’s exact test for categorical variables. Cox proportional-hazards regression was used to analyze the effect of bio-ADM and other variables on survival in uni- and multivariable analyses. Adjustments were performed for three different scores of disease severity: SOFA score [24], CardShock risk score [25], and IABP-SHOCK II risk score [26]. Biomarker data were log-transformed. For illustration, survival curves were drawn by using the Kaplan-Meier method. The dichotomization of patients was based on median bio-ADM level at inclusion, which was similar to the optimal cutoff estimated by analysis of the area under the curve of the receiver-operating characteristic (AUC ROC). Relationships between variables were assessed using Spearman correlation coefficient. A two-sided *p*-value < 0.05 was regarded as statistically significant. Statistical analyses were performed using R, version 3.5.1 (R Foundation for Statistical Computing, Vienna, Austria) with the statistical package ggplot2 for the data visualization.

## 3. Results

A total of 57 patients with CS were studied, including 21 patients who died within 30 days. Characteristics of the population at inclusion between 30-day survivors and non-survivors are presented in Table 1. In brief, non-survivors at 30 days compared to survivors at 30 days were of a higher age (76 (66–81) years old vs. 65 (53–73) years old, *p* = 0.008), were more likely to be female (52% vs. 22%, *p* = 0.040), had increased history of hypertension (43% vs. 14%, *p* = 0.024), and had a higher CardShock risk score (5 (4–7) vs. 4 (3–5), *p* = 0.044), lower mixed venous oxygen saturation (60 (50–70) % vs. 74 (68–81) %, *p* <0.001), and higher N-terminal pro-B-type natriuretic peptide (NT-proBNP) (6635 (1346–22,458) pg/mL vs. 1739 (566–5344) pg/mL, *p* = 0.017). 

Figure 1 shows that bio-ADM values were higher in 30-day non-survivors than 30-day survivors at inclusion (67.0 (54.6–142.9) pg/mL vs. 38.7 (23.8–63.6) pg/mL, *p* = 0.010), at 24 h (*p* = 0.012), and up to 48 h (*p* = 0.027). Of note, there was no influence of the vasopressor infused, epinephrine, or norepinephrine, on bio-ADM values at any time point (Appendix A). Characteristics of the population at inclusion according to the median of bio-ADM (bio-ADM = 53.8 pg/mL) are presented in Appendix A. In brief, patients in the high bio-ADM group were higher age (75 (67–81) year-old vs. 56 (52–67) year-old, *p* < 0.001), had higher CardShock risk score (5 (4–7) vs. 4 (3–4), *p* = 0.003), had lower diastolic arterial pressure (55 (46–61) mmHg vs. 60 (55–69) mmHg, *p* = 0.045), had lower estimated glomerular filtration rate (eGFR) (39.8 (31.1–49.0) mL/min/1.73 m^2^ vs. 69.5 (53.3–90.5) mL/min/1.73 m^2^, *p* < 0.001), had higher NT-proBNP (7543 (4427–20,766) pg/mL vs. 702 (375–1782) pg/mL, *p* < 0.001), and had higher occurrence of refractory CS (34% vs. 7%, *p* = 0.021). The 30-day all-cause risk of death was higher in patients with high values of bio-ADM (bio-ADM ≥ 53.8 pg/mL) at inclusion compared to those with low values of bio-ADM (bio-ADM < 53.8 pg/mL): unadjusted hazard ratio (HR) 3.90 (95% confidence interval [CI] 1.43–10.68, *p* = 0.008) (Figure 2). High bio-ADM remained associated with 30-day all-cause mortality after adjustment for catecholamine (epinephrine or norepinephrine) groups (adjusted HR 3.82 (95% CI 1.40–10.48, *p* = 0.009)). When the primary outcome was adjusted for the severity scores (SOFA score, CardShock risk score, and IABP-SHOCK II risk score), risks were found to be similar to those found in unadjusted analysis (Appendix A). The AUC ROC for the ability of bio-ADM value at inclusion to discriminate patients who died within 30 days from those who did not was 0.71 (95% CI 0.55–0.86). Using the Youden index, we identified an optimal cutoff of 54.6 pg/mL with the best sensitivity and specificity combination (sensitivity 76.2%, specificity 66.7%, positive predictive values 57.1%, negative predictive values 82.8%, respectively), which was similar to the median value of bio-ADM (53.8 mg/mL) at inclusion in the present study (Appendix A). When known predictors of mortality in CS were added to the bio-ADM model, NT-proBNP, eGFR, and lactate resulted in a slightly higher AUC for the prediction of 30-day all-cause mortality, although the differences were not statistically significant (Appendix A) [27,28,29].

Concerning secondary outcomes, patients with the occurrence of refractory CS had higher bio-ADM value at inclusion (90.7 (59.9–147.7) pg/mL vs. 40.7 (23.0–64.7) pg/mL *p* = 0.005), and patients with arrhythmias (i.e., comprised ventricular tachycardia, ventricular fibrillation, atrial fibrillation) had higher bio-ADM values at inclusion (63.7 (37.7–141.5) pg/mL vs. 42.8 (22.6–65.7) pg/mL *p* = 0.039), although there was no significant difference in bio-ADM value at inclusion between patients with and without MCS implementation (46.3 (36.2–108.0) pg/mL vs. 53.8 (26.7–85.2) pg/mL *p* = 0.74) (Figure 3). While epinephrine is associated with higher incidence of refractory CS in the princeps study [23], in the epinephrine group, patients with the occurrence of refractory CS had a higher bio-ADM value at inclusion compared to patients without the occurrence of refractory CS (90.7 (57.7–148.4) pg/mL vs. 39.8 (21.4–66.1) pg/mL *p* = 0.031) (Appendix A). Figure 4 shows a strong positive correlation between the value of bio-ADM at inclusion and the value of NT-proBNP at inclusion (r = 0.64, *p* < 0.001). Significant correlations were also observed between bio-ADM and pulmonary vascular resistance index (PVRI) (r = 0.49, *p* < 0.001), lactate (r = 0.31, *p* = 0.023), and eGFR (r = –0.47, *p* < 0.001). However, the bio-ADM values did not correlate with cardiac index (r = –0.21, *p* = 0.13), mean arterial pressure (r = –0.22, *p* = 0.16), systemic vascular resistance index (r = –0.25, *p* = 0.13), and high-sensitivity troponin T (r = 0.12, *p* = 0.38). 

6 and 12 patients had missing bio-ADM data at 24 h and 48 h, respectively. Of the patients for whom bio-ADM could not be measured at 24 h and 48 h, 3 patients had already died at 24 h. bio-ADM, bioactive adrenomedullin.

A base-10 log scale is used for the x-axis and the y-axis for the y axis of SVRI, PVRI, NT-proBNP, and hs-TnT. bio-ADM, bioactive adrenomedullin; CI, cardiac index; eGFR, estimated glomerular filtration rate; hs-TnT, high-sensitive troponin T; MAP, mean arterial pressure; NT-proBNP, N-terminal pro-B-type natriuretic peptide; PVRI, pulmonary vascular resistance index; SVRI, systemic vascular resistance index.

## 4. Discussion

The present study showed that high bio-ADM at ICU admission in patients with CS complicating AMI was associated with increased mortality at 30 days. There have been several studies investigating the prognostic impact of adrenomedullin in patients with CS [30,31,32]. However, the majority of these studies used a stable fragment of the precursor molecule, mid-regional pro-ADM (MR-proADM) [21]. The disadvantage of MR-proADM measurement is that it cannot distinguish between biologically active ADM and non-functional ADM. Recently the direct measurement of bio-ADM became available [21]. Our results using the data from the randomized trial study OptimaCC are in accordance with those of the observational study by Tolppanen et al., performed in 178 patients with CS [22]. The authors showed that high bio-ADM values at baseline and serially within 10 days were associated with high risk of death at 90 days; in addition, high bio-ADM was also related to impaired hemodynamics in CS patients. In the present study, a single measurement of bio-ADM after PCI was enough to detect the increased risk of worse outcome if bio-ADM measures were high at inclusion. Of note, cut-off values of bio-ADM to predict worse outcome were similar between our study and the study of Tolppanen et al. were around 55 pg/mL, and the AUCs for mortality prediction in both studies were acceptable for discrimination [22]. The present study had similar results to the study conducted by Tolppanen et al. on the prognostic impact of bio-ADM. On the other hand, the data on the correlation between bio-ADM and parameters of organ injury or functional impairment are novel. Bio-ADM was associated with some parameters of organ injury or functional impairment (PVRI, eGFR, NT-proBNP) and adverse events in the ICU in the present study. It is also notable that regarding the possibility that bio-ADM is sensitive to pulmonary vascular resistance, this could be explained by the fact that the lung is considered to be the primary site of ADM clearance [33], and bio-ADM is explicitly extracted from the pulmonary circulation [34,35].

On the other hand, CS is initiated by a reduction in cardiac output and subsequently affected by an activation of the renin–angiotensin–aldosterone system (RAS) [36] and endothelial dysfunction [37]. Indeed, ADM expression is also stimulated by neurohumoral factors and endothelial barrier leakage, and ADM is considered to be elevated to compensate for these factors [12,38,39]. Therefore, bio-ADM may be a useful biomarker for assessing the organ damage and severity associated with RAS stimulation and endothelial dysfunction. 

The most recognized functions of ADM are to stabilize the endothelial barrier and prevent vascular leakage, and to act on vascular smooth muscle cells, resulting in vasodilation. In fact, some studies have established the effects of exogenous administration of ADM in patients with heart failure and primary pulmonary hypertension [40,41]. In addition, on-going trials are assessing the benefits of anti-adrenomedullin antibody (Adrecizumab) on survival in the critically ill [42]. For instance, AdrenOSS-2 (NCT03085758) is assessing Adrecizumab in septic shock patients [43,44]. Furthermore, the on-going ACCOST-HH trial (NCT03989531) assesses whether Adrecizumab improves outcomes in patients with CS. 

## 5. Limitations

We acknowledge several limitations in this study. Firstly, the present observational analysis examines the association -not causation- between the bio-ADM and outcomes. Secondly, the results presented in this manuscript are a secondary analysis of the OptimaCC trial and should be regarded as hypothesis-generating, since the risk of confounding and bias can nott be excluded. Thirdly, a type II error cannot be excluded in some of the estimations made, especially in the outcomes with few events. In fact, some CIs estimated in our study were too wide, reflecting the risk of insufficient statistical power due to the limited sample size. Fourthly, this study was primarily composed of patients with CS complicating AMI. Therefore, it is unclear how the results will apply to the broader CS population. Finally, patients were from a single country, and external validation of our results should be demonstrated by further studies in other countries. 

## 6. Conclusions

The values of bio-ADM are associated with some parameters of organ injury and functional impairment in patients with CS complicating AMI and are prognostic for the occurrence of refractory CS and 30-day mortality.

## Figures and Tables

**Figure 1 jcm-10-04512-f001:**
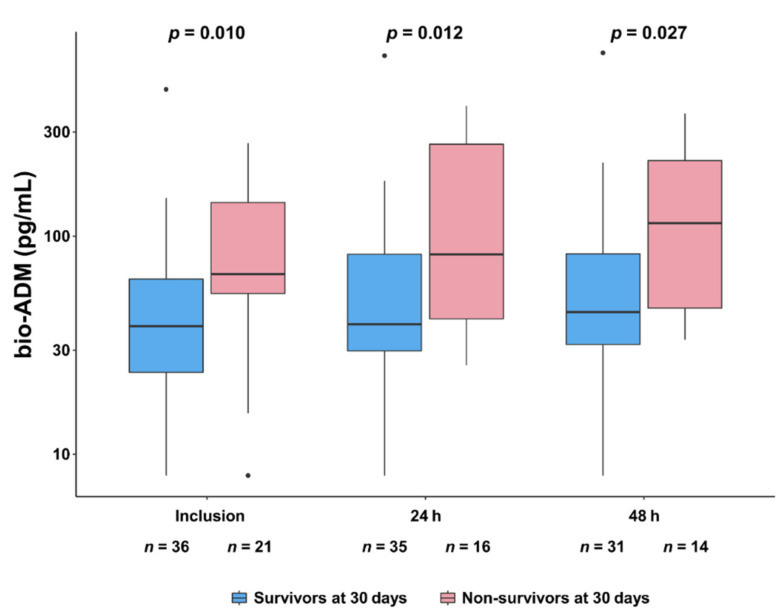
Comparison of time-course of bio-ADM, from inclusion to 48 h, between 30-day survivors and non-survivors. A base-10 log scale is used for the y-axis. Dots indicate outliers. Comparisons at each time point were performed using the Kruskal–Wallis test comparing bio-ADM values in 30-day survivors vs. 30-day non-survivors.

**Figure 2 jcm-10-04512-f002:**
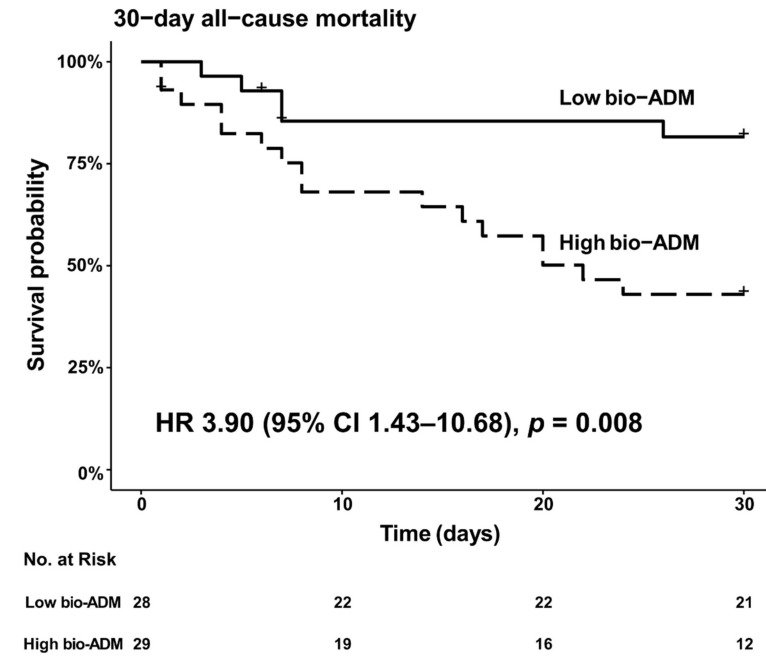
Kaplan–Meier analysis of 30-day all-cause mortality in cardiogenic shock patients with high bio-ADM value at inclusion (≥53.8 pg/mL) vs. low bio-ADM (<53.8 pg/mL). Dichotomization of patients was based on bio-ADM value 53.8 mg/mL, which was the median value of bio-ADM at inclusion (interquartile range 28.5–85.2 pg/mL). + indicates censoring. bio-ADM, bioactive adrenomedullin; HR, hazard ratio.

**Figure 3 jcm-10-04512-f003:**
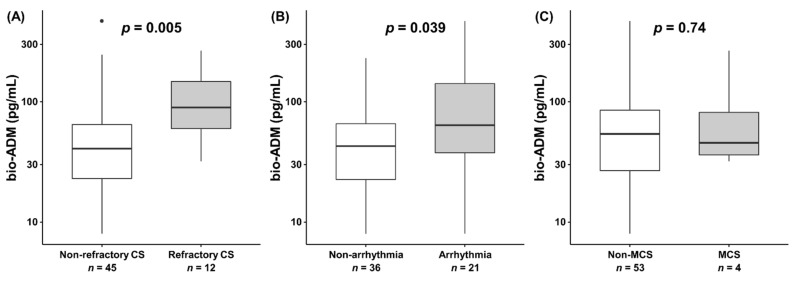
The value of bio-ADM at inclusion (boxplots) in (**A**) patients with or without refractory cardiogenic shock, (**B**) patients with or without arrhythmias (i.e., comprised ventricular tachycardia, ventricular fibrillation, atrial fibrillation), and (**C**) patients with or without mechanical circulatory support implementation. Dots indicate outliers. bio-ADM, bioactive adrenomedullin; CS, cardiogenic shock; MCS, mechanical circulatory support.

**Figure 4 jcm-10-04512-f004:**
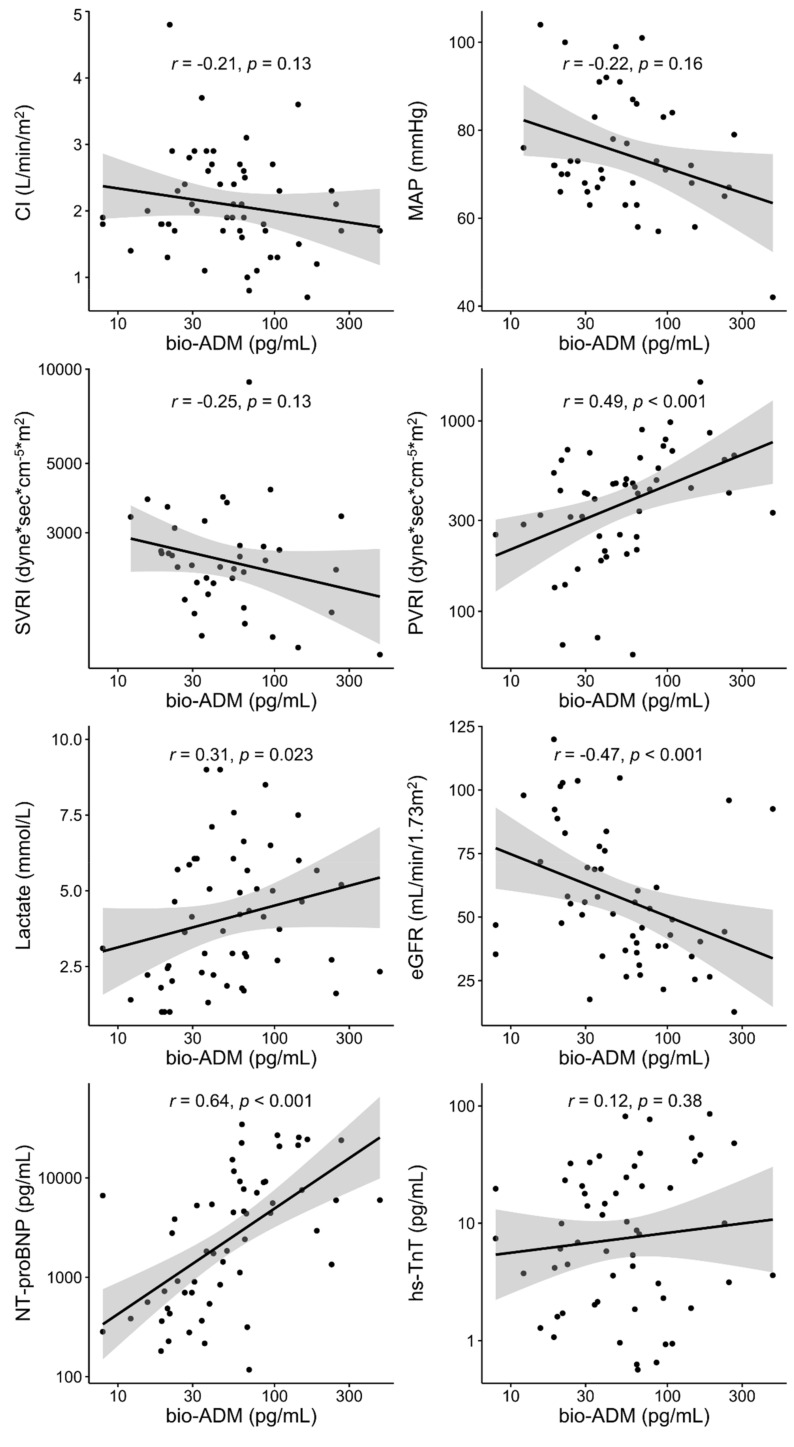
Spearman correlation between bio-ADM and hemodynamic and biological variables.

**Table 1 jcm-10-04512-t001:** Comparison of characteristics of cardiogenic shock patients between 30-day survivors and non-survivors.

Variables	Total *n* = 57	30-Day Survivors *n* = 36	30-Day Non-Survivors *n* = 21	*p*-Value
Demographics				
Age (years)	67 (55–77)	65 (53–73)	76 (66–81)	0.008
Female gender	19 (33%)	8 (22%)	11 (52%)	0.040
Medical history				
Hypertension	14 (25%)	5 (14%)	9 (43%)	0.024
Diabetes	6 (11%)	5 (14%)	1 (5%)	0.40
Stroke	4 (7%)	2 (6%)	2 (10%)	0.62
Myocardial infarction	4 (7%)	2 (6%)	2 (10%)	0.62
Severity scores				
SOFA score	9 (8–12)	9 (8–13)	10 (9–12)	0.61
CardShock risk score	4 (3–5)	4 (3–5)	5 (4–7)	0.044
IABP-SHOCK risk score	2 (1–3)	2 (1–3)	2 (1–4)	0.54
Clinical presentation at inclusion				
Body mass index (kg/m^2^)	25.4 (22.3–27.4)	25.4 (22.2–27.5)	25.7 (22.3–27.3)	0.93
Heart rate (bpm)	96 (81–111)	92 (77–111)	100 (88–107)	0.61
Systolic arterial pressure (mmHg)	101 (95–118)	100 (95–118)	102 (95–114)	1.00
Diastolic arterial pressure (mmHg)	58 (53–64)	58 (52–65)	56 (53–61)	0.70
Mean arterial pressure (mmHg)	72 (67–83)	72 (68–85)	68 (67–78)	0.47
SvO_2_ (%)	71 (60–79)	74 (68–81)	60 (50–70)	<0.001
Mechanical ventilation	46 (90%)	32 (89%)	14 (93%)	1.00
LVEF (%)	35 (25–40)	37 (30–44)	30 (24–38)	0.080
Laboratory findings at inclusion				
eGFR (mL/min/1.73 m^2^)	52.3 (38.2–76.5)	55.9 (38.6 – 83.7)	46.9 (34.6–59.2)	0.14
AST (UI/L)	471 (208–790)	430 (170–719)	558 (329–1175)	0.17
ALT (UI/L)	149 (74–224)	110 (56–198)	155 (99–298)	0.14
NT-proBNP (pg/mL)	2860 (666–7584)	1739 (566–5344)	6635 (1346–22,458)	0.017
hs-TnT (pg/mL)	7.7 (2.3–21.4)	5.8 (2.7–17.9)	11.8 (2.1–38.2)	0.24
Lactate (mmol/L)	3.9 (2.3–5.7)	3.9 (2.2–5.9)	3.9 (2.7–5.6)	0.81
Adverse events in ICU				
Refractory cardiogenic shock	12 (21%)	1 (3%)	11 (52%)	<0.001
Arrhythmias	21 (37%)	12 (33%)	9 (43%)	0.57
MCS implantation	4 (7%)	2 (6%)	2 (10%)	0.62

Data are presented as median (interquartile range) or *n* (%). Arrhythmias indicates ventricular tachycardia, ventricular fibrillation, or atrial fibrillation. AST, aspartate transaminase; ALT, alanine transaminase; eGFR, estimated glomerular filtration rate; hs-TnT, high-sensitive troponin T; IABP-SHOCK II, intra-aortic balloon pump in cardiogenic shock II; IQR, interquartile range; LVEF, left ventricular ejection fraction; MCS, mechanical circulatory support; NT-proBNP, N-terminal pro-B-type natriuretic peptide; SOFA, sequential organ failure assessment; SvO2, mixed venous oxygen saturation.

## Data Availability

The University Hospital Center in Nancy designed and sponsored OptimaCC trial. The executive committee can have unrestricted access to the data.

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
