# Peer review of "Elevated Plasma Bioactive Adrenomedullin and Mortality in Cardiogenic Shock: Results from the OptimaCC Trial"

_jcm, 2021, doi:10.3390/jcm10194512_

Round 1

Reviewer 1 Report

I read with interest the study by Takagi et al investigating into the predictive value of bioactive adrenomedullin for mortality in cardiogenic shock on grounds of acute myocardial infaction. I believe that the data will be of interest to readers. However, I believe that the manuscript can and should be improved prior to being published:

  • One major issue I see with the manuscript is that it does not get clear what the main objectives of the paper are as opposed to similar studies like the one by Tolppanen et al. Which parts of the results presented here represent a novelty and which parts are just confirmation of previously published data? This needs to be adequately discussed.
  • The methods section is extensive and should be shortened or partially transferred to the supplement. The randomization within the trial is of limited interest for the goal of the this analysis. 
  • If there is a correlation of known predictors of mortality in cardiogenic shock like GFR, Lactate and pro BNP. For the variables shown in Figure 4, it would be interesting to see bivariate models for each of the variables in combination with ADM to see whether ADM provides additional predictive information beyond these variables.
  • The discussion is way too short and superficial. I'd suggest to cite and refer to all studies examining adrenomedullin in cardiogenic shock or to state why this is not possible. A precise comparison of differences between this and existing studies is needed.
  • Many of the results remain undiscussed.
  • The authors conclude that "the values of bio-ADM reflect organ injury". This is an overstatement - bio-ADM is related to other biomarkers of organ injury, but no organs were scrutinized in the study. Would downtune here.
  • "An on-going trial assesses the benefits of the modulation of ADM pathway on mortality." That is no conclusion.
  • "Altogether, those data may confirm the importance of endothelial recovery in cardiogenic shock outcome." Is this really the conclusion to be drawn from the data shown in this manuscript?

Author Response

Response to Reviewer 1 Comments

Point 1: One major issue I see with the manuscript is that it does not get clear what the main objectives of the paper are as opposed to similar studies like the one by Tolppanen et al. Which parts of the results presented here represent a novelty and which parts are just confirmation of previously published data? This needs to be adequately discussed.

Response 1: We thank the reviewer for this suggestion. We have added sentences in the introduction and discussion sections.

However, the prognostic implication of bio-ADM and its relationship with parameters of organ injury or functional impairment in patients with CS remain poorly characterized.

The present study confirmed similar results to the study conducted by Tolppanen et al. on the prognostic impact of bio-ADM. On the other hand, the data on the correlation between bio-ADM and parameters of organ injury or functional impairment is novel.

Point 2: The methods section is extensive and should be shortened or partially transferred to the supplement. The randomization within the trial is of limited interest for the goal of this analysis.

Response 2:

We thank the reviewer for this suggestion. We have removed some sentences in the methods section.

Point 3: If there is a correlation of known predictors of mortality in cardiogenic shock like GFR, Lactate and pro BNP. For the variables shown in Figure 4, it would be interesting to see bivariate models for each of the variables in combination with ADM to see whether ADM provides additional predictive information beyond these variables.

Response 3:

We thank the reviewer for this suggestion. We have added in the result section:

When known predictors of mortality in cardiogenic shock were added to the bio-ADM model, NT-proBNP, eGFR, and lactate resulted in slightly higher AUC for the prediction of 30-day all-cause mortality, though the differences were not statistically significant (Table S2).

Point 4: The discussion is way too short and superficial. I'd suggest to cite and refer to all studies examining adrenomedullin in cardiogenic shock or to state why this is not possible. A precise comparison of differences between this and existing studies is needed.

Many of the results remain undiscussed.

Response 4:

We thank the reviewer for this suggestion. We have added in the discussion section:

There have been several studies investigating the prognostic impact of adrenomedullin in patients with CS. However, the majority of these studies used a stable fragment of the precursor molecule, mid-regional pro-ADM (MR-proADM). The disadvantage of MR-proADM measurement is that it cannot distinguish between biologically active ADM and non-functional ADM. Recently the direct measurement of bio-ADM became available.

Point 5: The authors conclude that "the values of bio-ADM reflect organ injury". This is an overstatement - bio-ADM is related to other biomarkers of organ injury, but no organs were scrutinized in the study. Would downtune here.

Response 5:

We thank the reviewer for this suggestion. We have downtuned in the abstract and conclusions section:

The values of bio-ADM are associated with some parameters of organ injury and functional impairment in patients with CS complicating AMI and are prognostic for the occurrence of refractory CS and 30-day mortality.

Point 6: "An on-going trial assesses the benefits of the modulation of ADM pathway on mortality." That is no conclusion.

"Altogether, those data may confirm the importance of endothelial recovery in cardiogenic shock outcome." Is this really the conclusion to be drawn from the data shown in this manuscript?

Response 6: We thank the reviewer for this suggestion. We have removed it in the conclusions section:

Reviewer 2 Report

Thank you for giving me the opportunity to review the original research paper entitled "Elevated plasma bioactive adrenomedullin and mortality in cardiogenic shock: results from the OptimaCC trial", submitted to JCM for consideration for publication. Takagi et al, examined -in a secondary analysis- the association between Bioactive adrenomedullin (bio-ADM) and organ injury, functional impairment, as well as survival in patients with cardiogenic shock (CS). The researchers found that Bio-ADM values at inclusion were correlated with pulmonary vascular resistance index, estimated glomerular filtration rate, and N-terminal pro-B-type natriuretic peptide and that increased Bio-ADM (above the median) was associated with increased risk (HR 3.90, 95% confidence interval 1.43–10.68, p=0.008) for 30-day all-cause mortality.

The authors should be congratulated on their attempt to examine "Novel" prognostic markers in patients with cardiogenic shock; a life-threatening condition. I have only some minor comments to make:

1) The authors should stress in the "Limitation section" that the current observational analysis examines the association -not causation- between the bio-ADM and outcomes. Therefore, as the authors state this is a "hypothesis generating" analysis since the risk of confounding and bias can't be excluded. On top of this, the number of patients is relatively small and the confidence intervals are relatively wide.

2) There is no established cut off value for Bio-ADM associated with worse outcomes, therefore the authors examined values above the median. Please comment on this. The reviewer thinks that this may be a fruitfull area of future research

3) The reviewer suggests that the authors create a table (In the Supplement) comparing the baseline characteristics between patients with high (above median) vs low (below median) bio-ADM and briefly discuss. 

4) Using the Youden index (line 153-155), the authors  reported an optimal cutoff (for the bio-ADM) of 54.6 pg/mL with the best sensitivity and specificity combination (sensitivity 76.2%, specificity 66.7%, respectively). Nevertheles, based on those values (sensitivity and specificity) the performance of this biomarker is relatively modest and this should be aknowledged by the authors in the discussion section. Please also report it's positive and negative predictive value

4) The authors state (lines 146-148) "Of note, there was no influence of the vasopressor infused, epinephrine, or norepinephrine, on bio-ADM values at any time point (Figure S1)". However, the graph depicts bio-ADM values the first 48 hours. The reviewer suggests that the 30 day mortality analysis (Figure 2) should be performed again adjusting for the vasopressor infused (epinephrine vs norepinephrine)

Author Response

Response to Reviewer 2 Comments

Point 1: The authors should stress in the "Limitation section" that the current observational analysis examines the association -not causation- between the bio-ADM and outcomes. Therefore, as the authors state this is a "hypothesis generating" analysis since the risk of confounding and bias can't be excluded. On top of this, the number of patients is relatively small and the confidence intervals are relatively wide.

Response 1: We thank the reviewer for this suggestion. We have added in the limitation section:

We acknowledge several limitations in this study. Firstly, the present observational analysis examines the association -not causation- between the bio-ADM and outcomes. Secondly, the results presented in this manuscript are a secondary analysis of the OptimaCC trial and should be regarded as hypothesis-generating since the risk of confounding and bias can't be excluded.

Point 2: There is no established cut off value for Bio-ADM associated with worse outcomes, therefore the authors examined values above the median. Please comment on this. The reviewer thinks that this may be a fruitful area of future research

Response 2: We thank the reviewer for this suggestion. We have added in the methods and result section:

Line 130:

For illustration, survival curves were drawn according to the median values by using the Kaplan-Meier method. Dichotomization of patients was based on meduan bio-ADM level at inclusion, which was similar to the optimal cutoff estimated by analysis of the area under the curve of the receiver-operating characteristic (AUC ROC).

Line 157:

Using the Youden index, we identified an optimal cutoff of 54.6 pg/mL with the best sensitivity and specificity combination (sensitivity 76.2%, specificity 66.7%, positive predictive values 57.1%, negative predictive values 82.8%, respectively), which was similar to the median value of bio-ADM (53.8 mg/mL) at inclusion in the present study (Figure S3).

Point 3: The reviewer suggests that the authors create a table (In the Supplement) comparing the baseline characteristics between patients with high (above median) vs low (below median) bio-ADM and briefly discuss.

Response 3: We thank the reviewer for this comment. We have added the new Table (Table S1)

entitled Comparison of characteristics of cardiogenic shock patients between high bio-ADM and low bio-ADM groups at inclusion. We also added result data in results section:

Characteristics of the population at inclusion according to the median of bio-ADM (bio-ADM = 53.8 pg/mL) are presented in Table S1. Briefly, patients in the high bio-ADM group had higher age (75 [67–81] year-old vs. 56 [52–67] year-old, p < 0.001), had higher CardShock risk score (5 [4 - 7] vs. 4 [3 - 4], p = 0.003), had lower diastolic arterial pressure (55 [46–61] mmHg vs. 60 [55–69] mmHg, p = 0.045), had lower estimated glomerular filtration rate (eGFR) (39.8 [31.1 - 49.0] mL/min/1.73m2 vs. 69.5 [53.3 - 90.5] mL/min/1.73m2), had higher NT-proBNP (7543 [4427 - 20766] pg/mL vs. 702 [375 - 1782] pg/mL, p < 0.001), and had higher occurrence of refractory CS (34 % vs. 7 %).

Point 4: Using the Youden index (line 153-155), the authors reported an optimal cutoff (for the bio-ADM) of 54.6 pg/mL with the best sensitivity and specificity combination (sensitivity 76.2%, specificity 66.7%, respectively). Nevertheless, based on those values (sensitivity and specificity) the performance of this biomarker is relatively modest and this should be acknowledged by the authors in the discussion section. Please also report it's positive and negative predictive value.

Response 4: We thank the reviewer for this suggestion. We have added in the result and discussion sections:

Line 157:

Using the Youden index, we identified an optimal cutoff of 54.6 pg/mL with the best sensitivity and specificity combination (sensitivity 76.2%, specificity 66.7%, positive predictive values 57.1%, negative predictive values 82.8%, respectively), which was similar to the median value of bio-ADM (53.8 mg/mL) at inclusion in the present study (Figure S3).

Line 188:

Of note, cut-off values of bio-ADM to predict worse outcome were similar between our study and the study of Tolppanen et al. were around 55 pg/mL, and the AUCs for mortality prediction in both studies were acceptable for discrimination.

Point 5: The authors state (lines 146-148) "Of note, there was no influence of the vasopressor infused, epinephrine, or norepinephrine, on bio-ADM values at any time point (Figure S1)". However, the graph depicts bio-ADM values the first 48 hours. The reviewer suggests that the 30-day mortality analysis (Figure 2) should be performed again adjusting for the vasopressor infused (epinephrine vs norepinephrine).

Response 5: We thank the reviewer for this suggestion. We have added in the result section:

High bio-ADM remained associated with 30-day all-cause mortality after adjustment for catecholamine (epinephrine and norepinephrine) groups [adjusted HR 3.82 (95% CI 1.40–10.48, p = 0.009)].